# Involvement of Non-Parental Caregivers in Obesity Prevention Interventions among 0–3-Year-Old Children: A Scoping Review

**DOI:** 10.3390/ijerph19084910

**Published:** 2022-04-18

**Authors:** Andrea Ramirez, Alison Tovar, Gretel Garcia, Tanya Nieri, Stephanie Hernandez, Myrna Sastre, Ann M. Cheney

**Affiliations:** 1Department of Nutrition and Food Sciences, University of Rhode Island, Kingston, RI 02881, USA; andrea_ramirez@uri.edu; 2Department of Behavioral and Social Sciences, Brown University School of Public Health, Providence, RI 02903, USA; alison_tovar@brown.edu; 3Graduate School of Education, University of California Riverside, Riverside, CA 92521, USA; gretelgarcia001@gmail.com; 4Department of Sociology, University of California Riverside, Riverside, CA 92521, USA; tanya.nieri@ucr.edu (T.N.); msast003@ucr.edu (M.S.); 5School of Public Policy, University of California Riverside, Riverside, CA 92507, USA; shern202@ucr.edu; 6Department of Social Medicine, Population and Public Health, School of Medicine, University of California Riverside, Riverside, CA 92521, USA

**Keywords:** child feeding, child growth, child weight, early childhood obesity, prevention interventions

## Abstract

Introduction: We examined the scope of literature including non-parental caregiver involvement in child obesity prevention interventions. Methods: We conducted a scoping review following the Arksey and O’Malley framework, including only studies reporting the effect of an intervention on growth, weight, or early childhood obesity risk among children ages 0 to three years, published between 2000 and 2021. Interventions that did not include non-parental caregivers (adults regularly involved in childcare other than parents) were excluded. Results: Of the 14 studies that met the inclusion criteria, all were published between 2013 and 2020, and most interventions (*n* = 9) were implemented in the United States. Eight of the 14 interventions purposefully included other non-parental caregivers: five included both parents and non-parental caregivers, and the remaining three included only non-parental caregivers. Most interventions (*n* = 9) showed no significant impact on anthropometric outcomes. All interventions found improvements in at least one behavioral outcome (e.g., food groups intake (*n* = 5), parental feeding practices (*n* = 3), and screen time (*n* = 2)). This review can inform future interventions that plan to involve non-parental caregivers, which may be beneficial in shaping early health behaviors and preventing obesity early in life.

## 1. Introduction

Childhood obesity is a serious public health concern [1]. Globally, 38.3 million children under the age of five experience overweight or obesity; and in the United States (U.S.), 13.4% of 2- to 5-year-old children have excess weight [2], with racial/ethnic minority children at greater risk [3]. Obesity in early life contributes to a critical public health burden due to hospitalization, treatment expenses, and negative social and economic outcomes [4,5]. The disparity observed at this young age presents an early window of opportunity to develop prevention interventions that foster healthy habits that can continue into adulthood [6]. Even though the first several years of life represent a critical period for health, development, and obesity prevention [7], few prevention programs intervene during these years [8,9,10,11,12,13].

Although care of a child early in life commonly involves a parent or guardian, often a mother, as the primary caregiver, other caregivers such as grandparents, extended family and household members, friends, or childcare providers may also provide care [14,15,16]. Non-parental caregivers are involved in child feeding by controlling availability and access of foods and can serve as active role models who can promote or hinder healthy feeding [17]. They also help shape other behaviors such as physical activity and screen use [18]. For example, evidence suggests that grandparents play an important role in child health behaviors and weight [19]. However, grandparents find it difficult to discuss childhood obesity, a situation that highlights the need to include grandparents in interventions [20]. Furthermore, parents may rely on some type of childcare (e.g., Early Head Start programs, daycare centers), and given that many children can spend a significant amount of time under the care of childcare providers early in life, it is critical to include these caregivers in prevention interventions [21,22,23,24]. 

Therefore, non-parental caregiver involvement has been identified as an important component in interventions intended to improve health behaviors [25,26]. However, few interventions aimed to prevent childhood obesity actively involve or target non-parental caregivers [27]. Most early-life obesity prevention interventions focus primarily on parents or a single caregiver in the immediate family and emphasize parent–child interactions, an approach that may not serve with parents in households where other non-parental caregivers are involved in feeding and care [28]. 

By intervening at an early age, the behaviors of parents and non-parental caregivers that can impact child diet can more easily be shaped. This is especially true for children with ages between 0 to three years, who are not yet in the school system and are often cared for by multiple caregivers. Understanding the state of research on childhood obesity prevention and the participation of non-parental caregivers in these interventions is critical and necessary to identify research gaps and opportunities. The goal of this review was to examine the scope of existing obesity prevention interventions among children 0 to three years, that involved, non-parental caregivers and to describe the non-parental caregivers’ characteristics and involvement in these interventions. 

## 2. Materials and Methods

### 2.1. Design

We conducted a scoping review of literature on obesity prevention interventions among children aged 0 to three years that involved non-parental caregivers to understand the available research given the nascent area. Specifically, we sought to examine the scope of research on interventions that reported outcomes on child growth, weight, or any other anthropometric related outcome, and to describe the non-parental caregivers and their involvement in the intervention. We defined non-parental caregivers as any person regularly involved in a child’s care such as family members, neighbors, babysitters, or childcare providers, as reported in the study. 

Following the Preferred Reporting Items for Systematic Reviews and Meta-Analyses (PRISMA) guidelines for scoping reviews [29] and the Arksey and O’Malley’s (2005) five-stage framework [30], we identified a research question, examined relevant studies, selected studies for inclusion, charted, and analyzed the data, and summarized the results.

### 2.2. Search Strategy

We searched the Cochrane Library and PubMed databases to identify studies using a standardized search string, which followed the patient/population, intervention, comparison, and outcomes (PICO) framework. These terms included medical subject headings and keywords. A medical education and clinical outreach librarian at a public research institute provided advice and direction during the search strategy development and literature review. Table 1 provides the string of search terms used.

We applied a publication date filter during the search, in which we only included articles published from January 2000 to December 2021. We performed the search on 30 March 2022. 

### 2.3. Study Selection

Studies were included if they were: (1) peer-reviewed; (2) published between 2000 and 2021, as there has been a significant growth of scientific publication related to infant feeding and growth after the year 2000; (3) conducted in English or Spanish; (4) reported on growth, weight, or any other anthropometric related outcomes such as body mass index (BMI) as primary or secondary outcomes; (5) included children in the age range 0 to three years; and (6) included non-parental caregivers. Domestic (i.e., U.S.) and international studies were eligible. Excluded literature included dissertations, protocol papers, published abstracts, unpublished work, literature reviews, and meta-analyses. 

We identified 3641 references in the databases. The first author exported the references to Endnote, removed duplicates, and screened article titles and abstracts for inclusion; and then performed a hand search. Three authors (AR, AMC, SH) assessed the full text of the remaining articles against the eligibility criteria, and independently reviewed and extracted the data. AR and AMC reviewed the final set of articles, confirmed their inclusion, discussed uncertainties and ambiguities, and reviewed for non-parental caregiver participation in the intervention. 

### 2.4. Data Analysis

During the data extraction, eligible studies were imported into an Excel spreadsheet and organized by article citation, research question (s), involvement of other caregivers (yes/no), approach or theoretical model, use of community-engaged approaches, research design, type of research, data collection methods, key variables, scales or measures, sample size, participant age range, participant inclusion and exclusion criteria, main results, and limitations. We excluded articles that did not include non-parental caregivers as well as those articles that did not clearly or explicitly indicate non-parental caregiver status. 

Following recommendations for conducting scoping reviews, our analysis focused on describing the interventions, and non-parental caregivers’ characteristics. Based on this information, we identified gaps in the research and next steps for the involvement of non-parental caregivers in obesity prevention interventions [29,30,31]. 

## 3. Results

### 3.1. Study Selection

The search from the Cochrane Library and PubMed yielded 3641 articles. After removing duplicates, 2920 titles and abstracts were screened, of which 2754 were excluded from full review for not meeting the inclusion criteria (*n* = 1647) including not focusing on the relevant outcomes for this analysis (i.e., infant growth, weight, BMI), or the children not being within the age range. 

After the full-text review of 178 articles, 164 articles did not meet the inclusion criteria and were excluded (125 of which did not include non-parental caregivers). A final sample of 14 articles were included in the scoping review. The complete screening process is presented in Figure 1.

### 3.2. Study Characteristics

A summary of the studies’ characteristics is presented in Table 2. The articles included were published between 2014 and 2020. Most interventions were implemented in the U.S. (*n* = 10) [32,33,34,35,36,37,38,39,40,41] and Australia (*n* = 2) [42,43], and the remaining interventions were conducted in Turkey [44] and the Netherlands [45]. More than half of the interventions (*n* = 8) were randomized controlled trials (RCTs) [33,34,36,38,40,41,42,44], five were cluster RCTs [32,35,37,39,43], and one was a quasi-experimental study [45]. 

Sample sizes ranged from 42 adult–child dyads [35] to 450 adult–child dyads [36], with other studies including more than 500 children from childcare centers [37,39,41,43]. Nine studies targeted children two to five years [32,33,35,36,37,38,40,41,42], two studies extended this range to six years [43,44], one study included children between two to four years [45], and another study included children between 1.5 to four years [39]. One intervention was initiated during pregnancy (*n* = 1) [34]. No study included children within the specific age range 0 to three years. 

Thirteen interventions evaluated the intervention effects on both anthropometric and behavioral outcomes; the remaining intervention analyzed the effects on anthropometric outcomes only [34]. The majority of the interventions (*n* = 11) measured changes in anthropometrics as a primary outcome [32,33,34,35,36,37,38,40,41,42,45]; the remaining (*n* = 3) included changes in anthropometric measures as a secondary outcome [39,43,44]. 

Other primary outcomes included in these interventions were child physical activity (*n* = 4) [36,39,40,45], child dietary intake (*n* = 4) [36,40,41,43], child sleep (*n* = 3) [36,38,42], child screen time (*n* = 2) [36,42], and child diet quality (*n* = 1) [39]. Frequent secondary outcomes were related to the children’s diet, such as child dietary intake (*n* = 4) [32,33,38,42], and diet quality (*n* = 1) [43], physical activity (*n* = 4) [35,37,38,42], and screen time (*n* = 3) [33,38,42]. Other secondary outcomes included home or childcare environment (*n* = 3) [36,37,39], parental feeding practices (*n* = 3) [35,38,42], adult physical activity (*n* = 3) [35,36,39], adult stress (*n* = 1) [36], and adult self-perception of efficacy for health-related behaviors (*n* = 1) [33]. 

### 3.3. Description of the Intervention

Table 3 presents descriptions of each intervention in the scoping review. 

#### 3.3.1. Theoretical Framework

Eleven studies reported using a conceptual model, theoretical framework, or approach (e.g., community engagement): social cognitive theory (*n* = 4) [39,41,42,44], social ecological model (*n* = 3) [32,40,45], community-based participatory research (*n* = 2) [33,36], theoretical domains framework (*n* = 1) [43], the social contextual framework (*n* = 1) [38], and self-determination theory (*n* = 1) [39].

#### 3.3.2. Intervention Duration

Intervention duration was on average 8.9 ± 7.5 months with a range from six weeks [44] to two years [33]. Approximately half of the interventions lasted less than three months (*n* = 6) [32,34,35,38,42,44], followed by six months (*n* = 1) [40], nine months (*n* = 1) [39], one year (*n* = 5) [36,37,40,43,45], and more than one year (*n* = 1) [33]. 

#### 3.3.3. Sample Characteristics

Of the ten studies reporting the race or ethnicity of the participants [32,33,34,35,36,38,39,40,41,42], the majority (*n* = 9) [32,33,34,35,36,38,39,40,41] included primarily racial/ethnic minorities, with Latinx/Hispanic, Black/African American, and American Indigenous (AI) being the most common groups. In five of these eight studies, the intervention was aimed at specific racial/ethnic groups such as AI on tribal reservations [33,36], Latinx/Hispanic [40,41], and non-Hispanic Black women [34]. Of the studies that did not report the participants’ race/ethnicity (*n* = 4) [37,42,43,44,45], two partnered with institutions serving low-income children [37,45] and the remaining two focused on socioeconomically diverse communities [43,45]. 

### 3.4. Intervention Characteristics and Non-Parental Caregivers’ Description and Role in the Intervention

The interventions’ characteristics and non-parental caregivers’ description and role are presented in Table 3 and Table 4. Non-parental caregivers in the interventions included childcare staff (*n* = 7) [32,37,39,40,41,43,45], other family members (*n* = 5) [33,34,35,38,42], non-relatives (*n* = 1) [34], community organizations (*n* = 1) [45], and other unspecified caregivers (*n* = 6) [33,34,36,38,42,44]. Of the 14 interventions, six included the non-parental caregivers along with a parent [32,34,40,41,44,45]. 

#### 3.4.1. Interventions That Included Non-Parental Caregivers Post-Hoc

Six interventions [33,35,36,38,42,44] did not initially aim their intervention activities toward non-parental caregivers; however, these caregivers were included because they were involved in the participating child’s care. 

Three of the six interventions targeted parents as the primary caregivers [35,38,42]; but did not exclude alternative non-parental caregivers. More than 90% of the participants in these three studies were the children’s mother, and non-parental caregivers included biological fathers (<5%) [35,38,42], adoptive mothers (2%) [35], stepmothers (1.8%) [38], grandmothers (2%) [35], and other unspecified caregivers [38,42]. 

Two interventions of the six that involved non-parental caregivers post-hoc, two were conducted with AI participants [33,36] and included any caregivers as primary adult participants. In one of these two studies, approximately 85% of the enrolled caregivers were the children’s mother, and the rest were the children’s grandparent/other (12.7%) and father (2%). In the other study, the caregiver–child relationship was not described, though most of the caregivers were women (94.7%) [36]. 

In the remaining intervention of the six that involved non-parental caregivers post-hoc [44], non-parental caregivers were included after the intervention started, since the investigators recognized that participating parents dealt with time limitations. The investigators included home visits to caregivers who were involved in the children’s care, along with parents receiving the intervention at doctors’ offices. 

#### 3.4.2. Interventions That Purposely Included Non-Parental Caregivers

Eight interventions targeted non-parental caregivers. Of those, five included both parental and non-parental caregivers [32,34,40,41,45], and three included only non-parental caregivers [37,39,43].

These interventions included mainly childcare providers as non-parental caregivers (*n* = 7) [5,32,37,39,40,41,43]. The primary intervention components included training sessions where providers received tailored nutrition and physical activity education and support (*n* = 5) [37,39,40,41,45] strategies to change the physical or social environment (*n* = 5) [32,39,40,41,45], or the implementation of policies or standards (*n* = 5) [32,39,40,41,43]. Of these seven interventions involving childcare, four included a family component in which families received their own set of information and activities separately from the intervention the childcare staff received [32,45], and one intervention involved community organizations related to physical activity and nutrition [45]. 

The remaining intervention included pregnant woman and study partner dyads [34]. The former was asked to identify a person who was involved in the child’s care and was important in their decision-making about infant care. The study partners received their own set of educational material and participated in home sessions with the mothers. 

### 3.5. Intervention Settings

The interventions that purposely included non-parental caregivers were primarily implemented at childcare centers or early care education (ECE) settings (*n* = 7) [32,37,39,40,41,43,45] whereas most interventions that purposely did not include non-caregivers were implemented at home (*n* = 3) [33,36,42]. Only two of the 14 interventions included in this review were multi-setting: one was delivered in doctors’ offices together with home visits [44], and the other targeted the preschool, home, and community settings [45]. The intervention settings are described in Table 3.

### 3.6. Intervention Activities

In general, most interventions were conducted in person (*n* = 11) [32,33,34,35,37,38,39,40,41,44,45]. The remaining interventions were web-based with facilitators providing feedback (*n* = 2) [42,43] and a mail-delivered toolkit (*n* = 1) [36]. Two of the 12 studies [36,42] included a social media component. 

Most interventions, independent of caregiver involvement, provided participants with informational and educational material (e.g., CDs, books, newspaper articles, emails, or recipe cards) [32,33,34,36,37,38,40,41,42,43,44,45]. The intervention activities are described in Table 3.

### 3.7. Intervention Results

Most of the interventions (*n* = 9) showed non-significant improvements in children’s anthropometric outcomes. Of the five interventions that found positive anthropometric outcomes, three were childcare center based interventions with teacher, parent, and policy implementation components [32,40,41], one involved childcare staff and consisted of program integration into existing public health infrastructure [37], and the last one included a healthy lifestyle toolkit delivered by in-home mentoring (intervention) compared to receiving the material in the postal mail (control) for any adult caregivers that lived with the child [33]. The latter found results only when the control and the intervention participants were combined. Nonetheless, all 14 interventions had a positive impact on one or more behavioral outcomes, independent of the non-caregivers being primarily targeted. 

All of the interventions that targeted parental feeding practices (*n* = 3) [35,38,42], home or childcare environments (*n* = 3) [36,37,39], and adult self-perception of efficacy for health-related behaviors (*n* = 1) [33] found significant improvements. Most interventions found positive outcomes in child food or food group intake (*n* = 7 out of 10) [32,33,36,40,41,42,43], screen time (*n* = 2 out of 6) [33,44], or child diet quality (*n* = 1 out of 2) [39]. However, only a few interventions found significant improvements in child physical activity (*n* = 1 out of 9) [45], and parent physical activity (*n* = 1 out of 3) [36]. No interventions in this analysis found changes in child sleep [36,38,42] or adult stress [36]. All studies using qualitative methods presented positive results regarding acceptability, usefulness, or relevance of interventions [33,35,36]. 

## 4. Discussion

This scoping review found that there are very few obesity prevention interventions for children 0 to three years that include non-parental caregivers such as other family members or childcare providers. Although we found 125 articles that focused on interventions addressing child growth, height, BMI, or early childhood obesity risk, only 14 included non-parental caregivers. Despite the recognition of non-parental caregivers’ involvement in childcare, we found that interventions continue to target primarily parents (specifically mothers). Of the 14 interventions that were analyzed in this review, only eight interventions included non-parental caregivers as part of the intervention design, whereas the remaining six interventions included non-parental caregivers later in the study in a post-hoc manner, as recruitment was flexible to include non-parental caregivers actively involved in the child’s care. These findings show that when interventions are being developed, non-parental caregivers, who might have an important influence on children’s health, are being overlooked.

Given the important role non-parental caregivers play in shaping health behaviors early in life, there is a need to develop interventions that actively involve them. By not doing so, prevention efforts may fail to capture the experiences of families with caregiving arrangements that do not rely solely or primarily on mothers. Families involving single parents, multi-generation, low-income, or same-sex parents as well as families that reside in multi-family households and low-income families could benefit from prevention efforts that address non-parental caregivers. Additionally, including non-parental caregivers in interventions along with parents can provide peer support; this can be particularly important among low-income families that bear the burden of social and economic factors on health outcomes [34,46]. 

Wasser et al. (2020) suggest that the definition of non-parental caregivers should extend beyond traditional notions of childcare involvement (e.g., grandmothers) and that interventions should consider the mothers’ circumstances (e.g., single parent, full-time employee) when considering other non-parental caregivers in infant care and feeding [34]. 

Childcare providers were also identified as non-parent caregivers in four of the studies presented in this analysis. Childcare has been identified as an important setting for intervention programs due to the considerable amount of time that children spend in childcare and the association between the feeding behaviors of childcare providers and childhood obesity [21,22,23,47,48,49].

Most of the interventions included in this review did not find significant changes in anthropometric outcomes. This finding is similar to previous literature among children ages 0 to five whereby interventions that focus on diet or physical activity have not significantly impacted these outcomes [8]. These results may be due to the length of time between the intervention and the measurement of the outcomes [45], food insecurity [50], or short duration of the interventions [45], which would suggest that longer-term intervention may be needed. In this scoping review, almost half of the studies lasted less than three months [32,34,35,38,42,44], even though it is recommended that interventions for obesity prevention range between two to 12 months in length [51]. Interventions can also address responsive parental feeding practices, since they have been associated with reduced growth-related indicators of obesity risk; this marks the importance of not only what is being fed, but also how and when feeding occurs [52,53].

Nonetheless, all interventions in this review were effective in improving other behavioral outcomes. These findings suggest that interventions may be effective in parent and non-parent caregiver behavior change in the context of childcare specific to child screen time, dietary intake, physical activity, sleep, stress, and parental feeding practices to improve anthropometric outcomes. 

This review also points to the multiple environmental and social influences on childcare and the need for obesity prevention interventions to include community or environment-based strategies [54], especially among low-income communities [55]. Most studies in the review included racial/ethnic minority participants, which is incredibly valuable given the disproportionate burden of childhood obesity among racial/ethnic minority child populations [56]. However, attention to environmental and social influences on caregiving was limited; only one intervention in our review included community support for healthy behavior change among families [45]. In addition, interventions tend to focus on single settings and lack comprehensiveness [45]. Multi-setting interventions (e.g., the combination of childcare center and home environments) have shown more significant and beneficial results on weight-related outcomes compared to single-setting interventions [49]; however, in this scoping review, only two interventions included more than one setting [44,45]. As one study demonstrated, including both the preschool and home setting decreased sedentary behavior more than focusing only on the preschool setting, which implies that children benefit if there are also changes in the home setting [45]. This underscores the importance of targeting multiple environments, in particular, the home and preschool, as previously suggested [57]. 

One way this can be conducted is by employing principles of community-based participatory research (CBPR) in health research, which privileges the needs of patients, community members, and diverse stakeholders in research [58]. As we found in our review, two studies in engaged American Indian community members and other key stakeholders in decision making about the study design, incorporation of American Indian values and knowledge-based approaches, and development of intervention material [33,36]. This approach informed a more inclusive study design permitting non-parent caregivers to be recipients of the intervention. Another study [34] employed CBPR approaches and designed their research to be inclusive of the diversity of adults involved in infant care, which prompted mothers to invite the infant’s father, grandmother, other relatives (e.g., aunts, siblings, cousins), or nonrelative adults. Similarly, another Such approaches demonstrate the value of CBPR and the importance of community and key stakeholder input into studies designed to address health disparities in early childhood obesity [59]. 

Several limitations of our review should be considered. First, articles in languages other than English or Spanish were not included in our analysis; this exclusion limits the generalizability of our findings to other countries. Second, although we conducted a systematic search in the database for broad coverage, our search strategy may not have identified all existing childhood obesity prevention interventions. Third, we selected the age range of 0 to three to capture early life childcare and feeding prior to the start of preschool, according to the definition of the Center for Disease Control [60]. Because of this focus, we included studies involving preschool age children only if the intervention targeted early life (i.e., 0 to three years), which overlaps with preschool years. It is possible that some articles that include interventions with two- to three-year-old children may have been excluded from our search. Fourth, studies in which non-parental caregivers’ participation was unclear or unspecified may have been excluded from the analysis. Fifth, in some of the studies included in this review [33,35,36,38,42], it was not clear whether the non-parental caregivers were substitutes for the parents as the primary caregiving parent was unavailable, or if the non-parental caregivers were the primary caregiver. This distinction could help explore the implications for interventions where multiple caregivers belong to the same family or live in the same household as well as the dynamics between multiple caregivers and how care is managed among them. In addition, information about the non-parental caregivers was not described extensively in some studies, which was a limitation to describing their characteristics. Sixth, we only included interventions that included anthropometric measures as outcomes; some obesity prevention programs therefore might have been overlooked. Finally, the included studies were heterogeneous, and we did not evaluate the quality of the studies; therefore, our results are subject to bias due to the internal and external validity of the study results. 

## 5. Conclusions

The scoping review provided a comprehensive overview of the state of research on early life obesity prevention interventions involving non-parental caregivers. Even though non-parental caregivers are commonly involved in the care of infants and toddlers, we found in this scoping review that only a few obesity prevention interventions for children between 0 and three years involved non-parental caregivers. The findings from this review highlight the need to target non-parental caregivers as participants in childhood obesity prevention interventions to better capture the multiple perspectives and competing interests that inform infant feeding practices and risk for early childhood obesity. In addition, this review can help inform the development of childhood obesity prevention interventions. The findings highlight the value of community-engaged approaches that can incorporate cultural values and knowledge-based approaches into interventions designed to address health disparities in early childhood obesity. For interventions to respond to the characteristics and needs of families, future research needs to better understand the role of non-parental caregivers and their dynamic with parents in infant and toddler feeding and early childhood obesity risk. 

## Figures and Tables

**Figure 1 ijerph-19-04910-f001:**
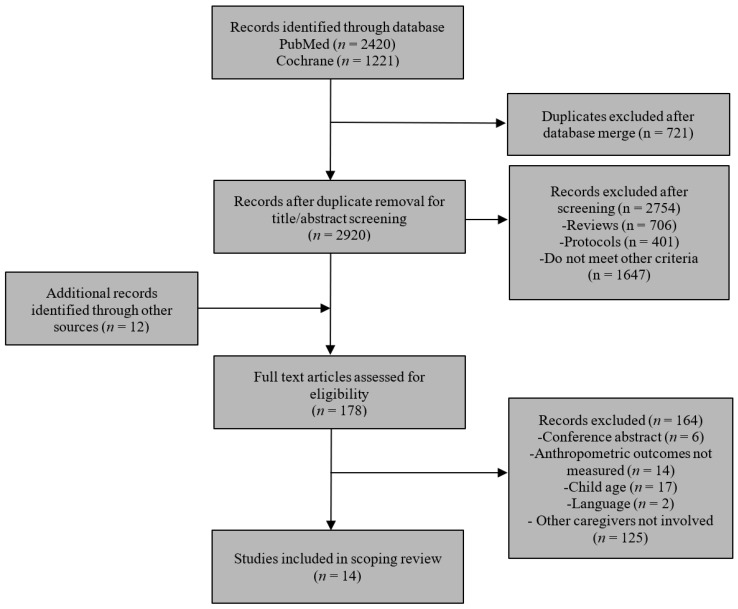
Screening process.

**Table 1 ijerph-19-04910-t001:** Search strategy.

Component	Search String
Population	(infant) OR (child, preschool) OR (infant, newborn) AND (caregivers) OR (“non-parent caregivers”) OR (“non parental”) OR (sibling) OR (“sibling caregivers”) OR (grandparent) OR (“grandparent caregivers”) OR (family) OR (“family caregivers”) OR (caretakers) OR (“secondary caregivers”) OR (parents) OR (father) OR (mother) OR (“non-maternal caregivers”) OR (“non-parental caregivers”)
Intervention	(“nutrition intervention”) OR (“nutrition education”) OR (“obesity interventions”) OR (“nutritional intervention”) OR (“obesity education”) OR (“obesity prevention education”) OR (“obesity prevention”) OR (“infant nutrition education”) OR (“infant feeding”) OR (“parent feeding practices”)) OR (“parental feeding practices”) OR (child nutrition sciences) OR (infant nutritional physiological phenomena) OR (health education)
Outcome	(obesity) OR (“infant growth”) OR (pediatric obesity) OR (nutritional status)

**Table 2 ijerph-19-04910-t002:** Summary of studies evaluating obesity prevention interventions among children between 0 and three years that included non-parental caregivers.

Authors	Country	Research Question	Study Design	Sample Size	Sample Design	Infant Age
Natale et al. (2014)	U.S.	Is an obesity prevention intervention conducted in CCs effective in improving children’s diet, PA and weight?	RCT	307 adult—child dyads, teachers in 8 CCs	CCs that served low-income, ethnically diverse families, and the center teachers.	2–5 years
Yilmaz et al. (2015)	Turkey	Does an intervention applied at the health maintenance visit reduce screen time, meals eaten in front of television, child’s aggressive behaviors and improve BMI z-score?	RCT	363 adult–child dyads	Parents who brought their children to the well-childcare visits in a hospital. No inclusion or exclusion criteria were stated in the paper.	2–6 years
Tomayko et al. (2016)	U.S.	Is an obesity prevention toolkit more effective when delivered in-home vs. mailed to impact child and adult weight status, nutrition and PA behaviors, and self-efficacy?	RCT	150 adult–child dyads	Children 2–5 years old who lived with one primary caregiver; without major physical or behavioral conditions.	2–5 years
Haines et al. (2016)	U.S.	Does an obesity prevention intervention, embedded within a parenting program, result in smaller increases in children’s BMI and improves weight-related behaviors?	RCT	112 adult–child dyads	Excluded parents unable to respond to interviews in English or Spanish, and children with severe health conditions.	2–5 years
Stookey et al. (2017)	U.S.	Does the incorporation of Happy Apple Program into routine public health nursing services improve the nutrition and PA best practices in childcare, child BMI percentile and z-score?	Cluster RCT	43 CCs; 791–945 children annually	CCs that participated in CCHP nutrition screenings in 2011–2012 except those with funding from Head Start, the school or community college district.	2–5 years
Natale et al. (2017)	U.S.	Does an obesity prevention intervention conducted in childcare with ethnically diverse children improve child weight and diet?	RCT	1211 children in 28 CC	28 CCs that served low-income, ethnically diverse families, and teachers.	2–5 years
Tomayko et al. (2019)	U.S.	Does a healthy lifestyle promotion/obesity prevention program improve health behaviors, and weight status in AI children?	RCT	450 adult–child dyads	Adults with a dependent child from 4 tribal reservations and 1 urban site; with a cell phone; they were not required to be AI, and the adult did not have to be the biological parent of the child.	2–5 years
Jastreboff et al. (2018)	U.S.	Does a novel mindfulness-based parent stress intervention decrease risk of early childhood obesity in low-income families?	Cluster RCT	42 adult–child dyads	Low-income parents, with a child between 2–5 years-old, with high levels of perceived stress.	2–5 years
Hammersley et al. (2019)	Australia	Does a parent-focused, internet-based lifestyle program aimed to overweight children or at risk to becoming overweight, improve child BMI, obesity-related behaviors, parent modeling and self-efficacy?	RCT	86 adult–child dyads	Children were at or above the WHO fiftieth percentile for BMI for their age and sex; they were excluded if they had a medical condition that affected weight.	2–5 years
Grummon et al., 2019)	U.S.	Does a multi-pronged pilot intervention promoting healthier beverage consumption improve children’s beverage consumption and weight status?	Cluster RCT	154 adult–child dyads and staff from 4 CCs	Licensed CCs that participated in CACFP and served English or Spanish-speaking families.	2–5 years
Van de Kolk et al. (2019)	The Netherlands	Does a comprehensive intervention embed in ECE improve child PA, sedentary behavior, and BMI z-score?	Quasi-experimental	191 parent–child dyads from 21 preschools	Preschools in neighborhoods with low SES were eligible, with Dutch speaking parents.	2–4 years
Wasser et al. (2020)	U.S.	Does a home-based intervention for NHB women and their study partners improve infant size and growth?	RCT	430 women their study partner and child	NBH, pregnant women, who spoke English, between 18 and 39 years, with a singleton pregnancy.	0–12 months
Yoong et al. (2020)	Australia	Does a web-based menu-planning tool program that implements dietary guidelines in CCs improve children’s diet, BMI z scores and child health-related quality of life?	Cluster RCT	483 children in 35 CCs	CCs were responsible of the menu planning decisions with a menu planner, provided at least 1 meal and 2 snacks to children.	2–6 years
Ward et al. (2020)	U.S.	Does a FCCH-based intervention improve children’s diet and PA?	Cluster RCT	496 children in 166 FCCH	Convenience sample of FCCHs, that provided at least one meal and one snack.	1.5–4 years

PA = physical activity; RCT = randomized controlled trial, CCs = childcare centers; BMI = body mass index; CCHP = child care health program; AI = American Indigenous; NBH = Non-Hispanic Black; WHO = World Health Organization; CACFP = child and adult care food program; ECE = early care and education; SES = socio economic status; FCCHs = family child care homes.

**Table 3 ijerph-19-04910-t003:** Description of the interventions evaluating obesity prevention interventions among children between 0 and three years that included non-parental caregivers.

Authors/Intervention Name	Framework or Theoretical Model	Intervention	Sample Characteristics	Main Results
Natale et al. (2014) Healthy Inside—Healthy Outside (HI-HO)	SEM	6-month, culturally appropriate intervention that implemented a nutrition and PA curriculum. CCs teachers received two trainings about healthy menus, and child nutrition; parents received information about nutrition and PA, and at home activities once a month by a registered dietitian; the centers incorporated policies.	Majority of families identified as Hispanic, and 35% were Spanish-speaking. Providers were also ethnically diverse.	No significant changes in PA, weight z-score, height z-score; however, BMI z-score was negatively correlated with their participation in home activities, and 97% of the children with normal weight remained with normal weight. Children in the intervention consumed more F/V, 1% milk, and less juice and junk food.
Yilmaz et al. (2015) Name was not specified	SCT	Conducted by health care practitioners during maintenance visits in a hospital; non-parental caregivers received home visits. Consisted of four intervention components, with a total duration of 6 weeks. Included printed materials, interactive CDs, and a counseling call, which addressed consequences of increased screen time, and alternatives to watching TV.	Ethnicity and race were not presented in the results. Majority of families had an annual income between $10,000–20,000.	In the intervention group vs. control, there was a significant reduction in meals consumed in front of a screen, screen time of children, parents, and non-parental caregivers, and ins aggressive behaviors. There were no significant differences in BMI z-scores.
Tomayko et al. (2016) Healthy Children, Strong Families (HCSF)	CBPR	A two-year, family-based intervention, that included a healthy lifestyle toolkit delivered by in-home mentoring or by mail (control). Designed with AI community input. The 1st year included monthly home visits (60 min) by community mentors. Lessons targeted lifestyle behavioral changes: F/V, sweetened drinks, sweets intake, PA, and TV viewing. The 2nd year included newsletters and group meetings.	More than 90% of the participants were AI. Most families (75.9%) received WIC benefits.	No significant effect of the toolkit delivery method was found. When control and intervention groups were combined, there was a significant decrease in child BMI percentile and TV use, and an increase in F/V intake and adult-self efficacy. No change in adult BMI was observed.
Haines et al. (2016) Parents and TOTS Together	Social contextual framework	Family-based intervention, adapted from the Chicago Parent Program. Consisted of 9 weekly group sessions, 2 h each, delivered by trained facilitators, at a community health center. Included information about parental roles in promoting healthy nutrition, PA, and weight related behaviors. Parents were given educational materials to share with other caregivers.	Children were primarily Hispanic (59%) or Black/African American (22%). Majority of participants (87%) had annual household incomes at or be ≤$50,000.	No significant differences in BMI. Parents in the intervention vs. control decreased restrictive feeding practices relative. Similar changes in children’s weight-related behaviors were observed in the intervention and control parents.
Stookey et al. (2017) Healthy Apple Program (HAP)	Not reported	Consisted of the integration of HAP into CCHP. Delivered by CCHP public health nurses or health workers in CCs. The HAP portion adapted resources from the University of North Carolina Nutrition and Physical Activity Self-Assessment for Child Care program and provided CCs providers with PA and nutrition resources. Each CC received 16 h of training and support.	Providers at CCs that serve low-income children. Race-ethnic data were not available.	In the HAP + CCHP program, there was a significant increase in the proportion of children exposed to nutrition and PA best practices, and a significant reduction in child BMI.
Natale et al. (2017) Healthy Caregivers—Healthy Children (HC2)	SCT	12-month, CC-based, culturally sensitive obesity prevention, intervention aimed to low income, diverse children. Curriculums addressed nutrition and PA. The policy curriculum included the implementation of policies. The parent/teacher curriculum included joint bimonthly meetings and additional trainings. The child curriculum included weekly lessons and support for teachers from curriculum specialists.	Majority of families identified as Hispanic or African American and were Spanish-speaking. Providers were also ethnically diverse.	Children in intervention vs. control had a lesser increase of BMI percentile. No significant changes in F/V and unhealthy food consumption.
Tomayko et al. (2019) Healthy Children, Strong Families 2 (HCSF2)	CBPR	Home-delivered, aimed to improve health behaviors. For a year, monthly toolkits were sent to families, with information and supportive items (measuring cups, games). Social media consisted of 2 weekly text messages and a Facebook group.	Majority of participants (>84.9%) were AI/AN. All families, reported income <$20,000/year.	Significant improvements in adult and child diet patterns, adult F/V intake, adult PA and self-efficacy for health behavior, home nutrition environment. No significant changes in adult BMI or child BMI z-score, child PA, adult stress, adult/child sleep, and screen time.
Jastreboff et al. (2018) Name was not specified	Not reported	Behavioral intervention delivered by a therapist trained in mindfulness and cognitive behavior therapy. Consisted of 8 weekly group sessions with other parents and included nutrition and PA counseling, goal setting, stress reduction techniques, and mindful eating.	Majority of parents (71%) and children (63%) were low-income were non-white (63%).	In the control group, there was a significant increase in child BMI-percentile. Intervention vs. control participants significantly improved parental emotional eating ratings. No significant differences in PA.
Hammersley et al. (2019) Time2bHealthy	SCT	Web-based intervention, consisted of 6 modules (introduction, nutrition, PA, screen time, sleep) delivered for 11 weeks. Each module included reading materials, videos, etc. Dietitian provided feedback to improve their goals. Participants could join a Facebook group.	3.5% of the children in the study were Australian-Aboriginal.	No significant differences in the BMI, PA, screen time, or sleep outcomes. Intervention vs. control group showed a reduced intake of discretionary food, and parents improved self-efficacy and child feeding pressure to eat.
Grummon et al. (2019) Name was not specified	SEM	Multi-level,12- week intervention aimed to improve beverage intake. Delivered via CC by research assistants and childcare teachers. Targeted children, parents, and CC staff. Included environmental changes, implementation of rules and policies, and educational activities for parents. Children participated in activities at childcare.	Predominantly Hispanic/Latino and low-income, with two-thirds of parents reporting annual household income of $30,000 or less.	Children reduced their consumption of less-healthy beverages and increased their consumption of healthier beverages. Children’s likelihood of being overweight decreased by 3 percentage points.
Van de Kolk et al. (2019) SuperFIT	SEM and systems theory	12-mo implementation intervention to connect strategies between settings. Delivered by health promotion experts. The preschool component targeted the sociocultural environment, such as PA and nutrition practices of teachers and physical environment. The family component addressed the sociocultural, political, and economic environment. The community component aimed to increase connections between organizations involved in PA and nutrition. Children could participate in the preschool and family components (full intervention) or in the preschool component (partial intervention).	Childcare in low-income neighborhoods, families, caregivers, and teachers. Ethnicity/race information was not presented.	No significant changes in BMI z-score in overall PA levels. Sedentary behavior decreased more in the full intervention group.
Wasser et al. (2020) Mothers & Others	Not reported	Home-based, responsive feeding intervention delivered by trained peer educators. Consisted of 8 visits during pregnancy and after birth, an information toolkit, and four newsletters. Included information about breastfeeding, responsive feeding style, complementary feeding, TV/media, and infant sleep. Women selected a study partner to participate, and they received their own set of materials.	NHB mothers, the majority were low income. Data about partners were not presented.	No significant differences in infant growth.
Yoong et al. (2020) feedAustralia	Theoretical Domains Framework	Web-based, 12-month intervention designed to address barriers to guideline implementation. It included the use of a web-based menu-planning program, educational resources reminder to increase compliance), training, and support by health promotion officer).	CCs from different economic backgrounds. Ethnicity/race was not presented.	In the intervention group vs. control, there was a significant increase in mean child consumption of fruit and dairy and a reduction in consumption of discretionary foods. No significant differences were observed in diet quality, BMI z-scores, or child health-related quality of life.
Ward et al. (2020) Keys to a Healthy Family Child Care Homes	SCT and SDT	9-month intervention to improve children’s diet and PA. Delivered by health coaches through a workshop, a home visit, and telephone calls/emails. It addressed the children intrapersonal and interpersonal relations, and FCCH organizational level. The intervention included three modules (3 months each) regarding FCCH provider health, FCCH environment and FCCH business practice.	63.3% of the children and 74.1% of providers participating were African American. Half the providers had an associated degree or college credit.	Children in the intervention group significantly improved their diet quality, no changes were observed in BMI, BMI percentile and PA. Providers improved their diet quality and some components of the FCCH environment.

SEM = socio-ecological model; PA = physical activity; CCs = childcare centers; BMI = body mass index; SCT = social cognitive theory; CBPR = community-based participatory research; AI = American Indigenous; AN = American Native; F/V = fruit and vegetables; NBH = Non-Hispanic Black; WLZ = weight-for-length; WAZ = weight-for-age; LAZ = length-for-age; CCHP = child care health program; FCCH = family child care home; SDT = self-determination theory.

**Table 4 ijerph-19-04910-t004:** Role and characteristics of the non-parental caregivers in the interventions.

Authors/Intervention Name	Non-Parental Caregivers Included in the Intervention				Non-Parental Caregivers’ Characteristics
	Other Family Members	Childcare Staff	Other Non-Relatives	Others/Unspecified	
Natale et al. (2014) Healthy Inside—Healthy Outside (HI-HO)		X			Intervention was developed to include CC teachers in childcare centers with low-income, mainly Latino children.
Yilmaz et al. (2015) Name was not specified				X	Unspecified
Tomayko et al. (2016) Healthy Children, Strong Families (HCSF)	X			X	Intervention was developed to include any primary caregiver that lived with the child at home: 85% of the participating caregivers were the mother of the child, 2% were the father and 12.7% were the grandparent/other.
Haines et al. (2016) Parents and TOTS Together	X			X	Intervention was developed to include parents as the primary participants, but other adults participated: 92.9% were the mothers of the children, 5.4% were the fathers, and 1.8% were stepmother/other.
Stookey et al. (2017) Healthy Apple Program		X			Intervention was developed to include childcare providers. Unspecified providers’ characteristics
Natale et al. (2017) Healthy Caregivers—Healthy Children (HC2)		X			Intervention was developed to include CC teachers in childcare centers with low-income, mainly Latino children.
Tomayko et al. (2019) Healthy Children, Strong Families 2				X	Intervention was developed to include any adult primary caregiver and did not have to be the biological parent of the child. Adults in the intervention were on average 31.4 ± 7.8 years old, and they were mostly women (94.7%). Information about the relationship between the adults and the children was not provided.
Jastreboff et al. (2018) Name was not specified	X				Intervention was developed for parents as the primary participants, but other adults participated: 95% (n = 59) were biological mothers, one was the biological father, one was an adoptive mother, and one was a grandmother.
Hammersley et al. (2019) Time2bHealthy	X			X	Intervention was developed for parents as the primary participants, but other adults participated: 93% of the adults were the children’s mother, 5% were the father, and 2% were other.
Grummon et al. (2019) Name was not specified		X			Intervention was developed to include childcare providers. Unspecified providers’ characteristics
Van de Kolk et al. (2019) SuperFIT		X			Intervention was developed to include childcare providers. Unspecified providers’ characteristics
Wasser et al. (2020) Mothers & Others	X		X	X	Intervention included a study partner selected by the mother. Approximately, half of the study partners (54.6%) were the infant’s father, 27.5% were the infant’s grandmother, 11.5% were another type of relative (infant’s aunt, cousin, grandfather, sister or unspecified), and 6.4% were not non-relatives (mother’s roommate, infant’s godmother, or unspecified).
Yoong et al. (2020) feedAustralia		X			Intervention was developed to include childcare providers. Unspecified providers’ characteristics
Ward et al. (2020) Keys to a Healthy Family Child Care Homes		X			Intervention was developed to include providers in FCCH. Most of them were female, African American (74.1%).

FCCH = family child care home.

## Data Availability

The articles identified via this scoping review are available upon reasonable request from the corresponding author.

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
