# Peer review of "Involvement of Non-Parental Caregivers in Obesity Prevention Interventions among 0–3-Year-Old Children: A Scoping Review"

_ijerph, 2022, doi:10.3390/ijerph19084910_

Round 1

Reviewer 1 Report

This study is a review paper written for the purpose of examining the effects of obesity prevention interventions in non-parent children aged 0-3. It is confirmed that the study was systematically well written. Accordingly, my opinions agree to be published.
Edit it again to fit the journal format, and check simple spelling, spacing, and table size.

Author Response

Thank you for your positive feedback. We have edited the paper to fit the journal format as well as checked spelling, spacing, and table size.

Reviewer 2 Report

Comments:
Review of the paper
ijerph-1658827

This paper reviewed the intervention of non-parental caregivers on early childhood obesity.

After careful literature searching, I found that there was no relevant literature conducting in-depth review of this topic. This paper selects a lot of literatures as references, which have strong reliability.

However, the author's suggestions on the involvement of non-parental caregivers in obesity prevention interventions in the future were vague and general, and need to be more specific.

As a review, the deadline for selecting references in this manuscript is 2020, it is suggested that the authors should add some articles published in the last two years (2021, 2022).

To sum up, this review has a relatively novel perspective, authors should resubmit a revised version for reviewing again.

Author Response

After careful literature searching, I found that there was no relevant literature conducting in-depth review of this topic. This paper selects a lot of literatures as references, which have strong reliability.

Response: Thank you for the support of our scooping review and validating its important contribution to the literature.

The author's suggestions on the involvement of non-parental caregivers in obesity prevention interventions in the future were vague and general and need to be more specific.

Response: In the discussion section, we provided clear recommendations on ways that non-parental caregivers can be involved in obesity prevention interventions.

As a review, the deadline for selecting references in this manuscript is 2020, it is suggested that the authors should add some articles published in the last two years (2021, 2022).

Response: We extended the search to 2021 so as to include a complete year of review of additional articles. The search was thus from 2000 to 2021. While the search did not result in new articles published in 2021 for review, the revised search did result in two new articles (Natale et al., 204; 2017) in our review. We updated the results to reflect the updated review.

To sum up, this review has a relatively novel perspective, authors should resubmit a revised version for reviewing again.

Response: Thank you for your positive feedback. We believe the revised manuscript is improved.

Reviewer 3 Report

This review is examined the scope of literature including non-parental caregiver involvement in obesity prevention interventions among children 0 to 3 years.  The text of this manuscript  is divided into 5 paragraphs. The manuscript is written on the basis of 59 references. In my opinion this manuscript is not reader friendly and suffers from some flaws. But,  the issue of this article is interesting and important to analyze.

  1. The article is too long and chaotic.
  2. line 157 “No studies included children specifically from an age range 0 to 3 years” but your title, aim and conclusion are focused on 0–3-year-old children?
  3. A final sample of 12 articles were included in this scoping review.  The description of this articles  should be more compressive and shorter in the tables 2,3,4. The text in the tables should be more reader friendly. Please avoid to copy the excerpts from the cited references to the tables.
  4. The discussion section and main conclusion (line 406) should mainly focus on the interventions results.

Author Response

This review is examined the scope of literature including non-parental caregiver involvement in obesity prevention interventions among children 0 to 3 years. The text of this manuscript is divided into 5 paragraphs. The manuscript is written on the basis of 59 references. In my opinion this manuscript is not reader friendly and suffers from some flaws. But, the issue of this article is interesting and important to analyze.

Response: Thank you for your feedback. We revised the manuscript so that it is more reader friendly and easy to follow. We agree that the article itself is interesting and important.

The article is too long and chaotic.

Response: Throughout the manuscript, we edited the text and tables to reduce the text of the manuscript and make it easier to follow.

line 157 “No studies included children specifically from an age range 0 to 3 years” but your title, aim and conclusion are focused on 0–3-year-old children?

Response: We revised this statement to clarify that a single study did not focus on this age range, rather studies tended to focus on 0 to 2 or preschool age children (e.g., 3-4).

A final sample of 12 articles were included in this scoping review.  The description of this articles should be more compressive and shorter in the tables 2,3,4. The text in the tables should be more reader friendly. Please avoid to copy the excerpts from the cited references to the tables.

Response: We revised the tables for conciseness and clarity. For example, in Table 3, we omitted the column “outcome measures.” We believe the revised tables are easier to read. We also added two additional studies for a final sample of 14: 1) Natale et al. 2014, Is an obesity prevention intervention conducted in CCs effective in improving children’s diet, PA and weight? and 2) Natale et al., 2017 Does an obesity prevention intervention conducted in childcare with ethnically diverse children improve child weight and diet? which were identified in a revised research. We also revised to text, so the excerpts were concise and did not replicate text of the included references.

The discussion section and main conclusion (line 406) should mainly focus on the interventions results.

Response: We revised the discussion and conclusion section to focus on the intervention results. We reorganized the ideas so that discussion of the lack of literature on non-parental caregivers in early childhood obesity efforts is concentrated in the first several paragraphs.

Round 2

Reviewer 2 Report

Comments:
Review of the paper
ijerph-1658827-peer-review-v2

The authors have corrected the manuscript according to my suggestions. The article is now ready to be accepted and published.

Reviewer 3 Report

Thank you very much for a comprehensive response to my comments.